# Impact of the announcement and implementation of the UK Soft Drinks Industry Levy on sugar content, price, product size and number of available soft drinks in the UK, 2015-19: A controlled interrupted time series analysis

**Peter Scarborough**[1]\*, **Vyas Adhikari**[1], **Richard A. Harrington**[1], **Ahmed Elhussein**[2], **Adam Briggs**[1,3], **Mike Rayner**[1], **Jean Adams**[4], **Steven Cummins**[5], **Tarra Penney**[4], **Martin White**[4]

**1** Centre on Population Approaches for Non-Communicable Disease Prevention and Oxford Biomedical Research Centre, Nuffield Department of Population Health, University of Oxford, Big Data Institute, Headington, Oxford, United Kingdom, **2** Johns Hopkins University School of Medicine, Baltimore, Maryland, United States of America, **3** Warwick Medical School, The University of Warwick, Division of Health Sciences, Coventry, United Kingdom, **4** Centre for Diet & Activity Research, MRC Epidemiology Unit, University of Cambridge School of Clinical Medicine, Cambridge Biomedical Campus, Cambridge, United Kingdom, **5** Population Health Innovation Lab, Department of Public Health, Environments & Society, Faculty of Public Health & Policy, London School of Hygiene & Tropical Medicine, London, United Kingdom

\* peter.scarborough@ndph.ox.ac.uk

## Abstract

### Background

Dietary sugar, especially in liquid form, increases risk of dental caries, adiposity, and type 2 diabetes. The United Kingdom Soft Drinks Industry Levy (SDIL) was announced in March 2016 and implemented in April 2018 and charges manufacturers and importers at £0.24 per litre for drinks with over 8 g sugar per 100 mL (high levy category), £0.18 per litre for drinks with 5 to 8 g sugar per 100 mL (low levy category), and no charge for drinks with less than 5 g sugar per 100 mL (no levy category). Fruit juices and milk-based drinks are exempt. We measured the impact of the SDIL on price, product size, number of soft drinks on the marketplace, and the proportion of drinks over the lower levy threshold of 5 g sugar per 100 mL.

### Methods and findings

We analysed data on a total of 209,637 observations of soft drinks over 85 time points between September 2015 and February 2019, collected from the websites of the leading supermarkets in the UK. The data set was structured as a repeat cross-sectional study. We used controlled interrupted time series to assess the impact of the SDIL on changes in level and slope for the 4 outcome variables. Equivalent models were run for potentially levy-eligible drink categories ('intervention' drinks) and levy-exempt fruit juices and milk-based drinks ('control' drinks). Observed results were compared with counterfactual scenarios based on

**Data Availability Statement:** This study includes data collected from six online supermarkets (Tesco, Sainsbury's, Asda, Morrisons, Waitrose and Ocado). Data were collected by the researchers for the period November 2017 to February 2019. These were supplemented by data purchased from a commercial organisation (Brand View) for the period August 2015 to September 2018. These data were collated into five overlapping, but separate datasets for the purpose of analyses: ComparisonDB (for comparing trends in data collected by Brand View and by the researchers); ReformulationDB (for analysis of sugar levels over time); PriceDB (for analysis of price over time); SizeDB (for analysis of product size over time); NumberDB (for analysis of the number of products available over time). Please see Fig 1 for more details. Anonymised versions of the ReformulationDB, PriceDB, SizeDB and NumberDB datasets, where supermarket and product names are redacted and restricted to data collected by the researchers, are available from the Oxford University Research Archive (https://ora.ox.ac.uk/) under doi 10.5287/bodleian:v0Aq7PZqD. Because of legal restrictions to the use of copyrighted material it is not possible to share data openly which reveal the product or supermarket names, but unredacted versions of the dataset are available with a licensed agreement that they will be restricted to non-commercial use. Please contact the foodDB Data Access Committee (contact Trisha Gordon – trisha.gordon@ndph.ox.ac.uk or foodDBaccess@ndph.ox.ac.uk). The use of data collected by the researchers for this study is covered by the Intellectual Property Office's Exceptions to Copyright for Non-Commercial Research and Private Study (https://www.gov.uk/guidance/exceptions-to-copyright). The Brand View dataset purchased and used for the analyses in this paper can also be requested from the Data Access Committee under licence for the purpose of verification and replication of findings only. Further use of these datasets must be negotiated with the data owners Brand View (contact David Beech - info@ascentialedge.com).

**Funding:** This work was funded by grants from the UK National Institute for Health Research Public Health Research programme (numbers 16/49/01 and 16/130/01) for an evaluation of the UK Soft Drink Industry Levy, and by the NIHR Oxford Biomedical Research Centre (IS-BRC-1215-20008). PS is supported by a British Heart Foundation fellowship (FS/15/34/31656). MR is supported by the University of Oxford. JA and MW are funded by the University of Cambridge and Centre for Diet and Activity Research (CEDAR). CEDAR is UK Clinical Research Collaboration

extrapolation of pre-SDIL trends. We found that in February 2019, the proportion of intervention drinks over the lower levy sugar threshold had fallen by 33.8 percentage points (95% CI: 33.3–34.4, $p < 0.001$). The price of intervention drinks in the high levy category had risen by £0.075 (£0.037–0.115, $p < 0.001$) per litre—a 31% pass through rate—whilst prices of intervention drinks in the low levy category and no levy category had fallen and risen by smaller amounts, respectively. Whilst the product size of branded high levy and low levy drinks barely changed after implementation of the SDIL (−7 mL [−23 to 11 mL] and 16 mL [6–27ml], respectively), there were large changes to product size of own-brand drinks with an increase of 172 mL (133–214 mL) for high levy drinks and a decrease of 141 mL (111–170 mL) for low levy drinks. The number of available drinks that were in the high levy category when the SDIL was announced was reduced by 3 (−6 to 12) by the implementation of the SDIL. Equivalent models for control drinks provided little evidence of impact of the SDIL. These results are not sales weighted, so do not give an account of how sugar consumption from drinks may have changed over the time period.

## Conclusions

The results suggest that the SDIL incentivised many manufacturers to reduce sugar in soft drinks. Some of the cost of the levy to manufacturers and importers was passed on to consumers as higher prices but not always on targeted drinks. These changes could reduce population exposure to liquid sugars and associated health risks.

## Author summary

### Why was this study done?

- In March 2016, the United Kingdom Government announced the Soft Drinks Industry Levy (SDIL)—a tax on soft drinks that contain more than 5 g sugar per 100 mL. Fruit juices and milk-based drinks are exempt from the levy. The stated aim of the SDIL was to encourage the soft drinks industry to improve the healthiness of the drinks they produce, by reducing sugar content or reducing portion sizes. The SDIL was implemented in April 2018.

- This study measures the impact of the SDIL on the soft drinks that are available to buy in the UK to evaluate whether the SDIL achieved its aim of influencing industry practice.

### What did the researchers do and find?

- We used data on 209,637 observations of soft drinks available from UK supermarket websites at 85 time points between September 2015 and February 2019.

- At each time point, we measured the percentage of drinks with sugar levels greater than 5 g per 100 mL, the price of drinks, the volume at which they are sold, and the number of different drinks available to purchase and compared these with estimates of what would have happened if the SDIL was not introduced.

(UKCRC) Public Health Research Centre of Excellence. Funding for CEDAR from the British Heart Foundation, Cancer Research UK, Economic and Social Research Council, Medical Research Council, the National Institute for Health Research, and the Wellcome Trust, under the auspices of the UKCRC, is gratefully acknowledged. The views expressed are those of the authors and not necessarily those of the National Health Service, the NIHR, or the Department of Health and Social Care, UK. The funders had no role in study design, data collection and analysis, decision to publish, or preparation of the manuscript.

**Competing interests:** We have read the journal's policy and the authors of this manuscript have the following competing interests: ADMB and MW are members of the Faculty of Public Health, which has a position statement supporting a soft drink tax. AMDB is also a former member of the UK Health Forum, which had a position statement supporting soft drink taxes. ADMB has written various peer-reviewed and commissioned papers on soft drink taxes. MR is chair of Sustain: the alliance for better food and farming, which has for more than eight years campaigned for a sugary drinks tax in the UK. MW receives salary as Director of NIHR's Public Health Research Progamme, outside the submitted work.

**Abbreviations:** CITS, controlled interrupted time series; NHS, National Health Service; PHE, Public Health England; SDIL, Soft Drinks Industry Levy; SSB, sugar sweetened beverage.

- We found changes to sugar levels in drinks. The percentage of drinks with sugar over 5 g per 100 mL fell from an expected level of 49% to 15% over the time period. There was little change in the product size or the number of products available to consumers. The price of high sugar drinks increased after the implementation of the SDIL but only by one third of the amount of the tax.

## What do these findings mean?

- The results show that the SDIL was associated with a considerable impact on the soft drinks industry, particularly with regard to the amount of sugar in soft drinks. The SDIL was not associated with a reduction in the size of the soft drinks marketplace.

- These results are not weighted by sales of soft drinks, so we are not able to estimate the impact of these changes on sugar consumption.

## Introduction

Free sugars have been shown to be associated with obesity and type 2 diabetes [1,2], especially when consumed in liquid form [3,4]. Consumption of sugar sweetened beverages (SSBs) increases body weight in children [5,6] and has been associated with obesity [7,8], diabetes [9,10,11], hypertension [12], and cardiovascular disease [9,13] in adults. An estimated 3.6% of diabetes cases in the United Kingdom (and 8.7% of cases in the United States) are attributable to SSB consumption [14]—a condition that presently costs the National Health Service (NHS) around £10 billion a year [15].

In October 2015, in response to the Health Select Committee inquiry on Childhood Obesity [16], Public Health England published a report listing recommendations for reducing sugar consumption in children, including a tax on SSBs [17]. George Osborne, then Chancellor of Exchequer, announced in his budget of 16 March 2016 that the Government would introduce a UK Soft Drinks Industry Levy (SDIL) to be implemented on 6 April 2018 [18], allowing 2 years for manufacturers to prepare for the levy by reformulating drinks, reducing product sizes, or removing and/or introducing products from and/or to the marketplace. The SDIL is a levy on manufacturers and importers of soft drinks based on total sales of drinks aimed at influencing industry behaviour. This distinguishes it from most soft drink taxes introduced elsewhere [19], which are normally excise taxes, aimed at increasing price for the end consumer, with the intention of reducing demand for SSBs. To incentivise reformulation of sugar levels, the SDIL is a two-tiered levy: drinks over 8 g of sugar per 100 mL are levied at a rate of £0.24 per litre (higher levy tier); between 5 and 8 g of sugar per 100 mL, drinks are levied at a rate of £0.18 per litre (lower levy tier). Drinks with less than 5 g sugar per 100 mL are not levied (no levy tier) [20]. Soft drinks that are 100% fruit juice, at least 75% milk (or a milk replacement), contain greater than 1.2% alcohol (or are an alcoholic beverage replacement), or are produced or distributed by manufacturers and importers with UK sales less than 1 million litres per year are exempt from the SDIL, irrespective of sugar content. These rates were announced in March 2016 but not confirmed until 27 February 2017 in a prebudget statement. A more detailed description of the policy objectives for the SDIL can be found elsewhere [21].

Previous evaluations of soft drink taxes have focussed on their impact on price and consumer purchasing behaviour [22,23,24,25] but have not evaluated their impact on sugar content in drinks, product sizes, and product diversity within the marketplace. We hypothesised that the SDIL would have multiple impacts on the UK food and drink system [26], and here we report on the impact of the announcement (16 March 2016) and implementation (6 April 2018) of the SDIL on the proportion of soft drinks with sugar levels above levy thresholds, their price, the volume in which they are sold, and the number of soft drinks in supermarkets. We present results separately for 'branded' and 'own-brand' products (here we define 'own-brand' products as those manufactured and branded by supermarket and 'branded' products as all other drinks) because they occupy different places in the soft drinks marketplace. Consumers of own-brand products tend to be more motivated by price than by quality and perception of own-brands influence consumers' perception of the supermarket as a whole [27–28]. Manufacturers of branded and own-brand products therefore have different motivations and could react to the SDIL differently.

## Methods

### Outcome measures

Using a time-stamped data set of observations of soft drinks available in UK supermarkets between September 2015 and February 2019, we assessed whether the announcement and implementation of the SDIL had an impact on the following measures:

- The proportion of available drinks with sugar content greater than 5 g per 100 mL (the threshold over which the levy applies. An equivalent analysis considering the proportion of drinks with sugar content greater than or equal to 8 g per 100 mL—the higher levy threshold—is reported in S1 Appendix).

- The mean price (£ per 100 mL) of available soft drinks.

- The mean product size (mL) of available soft drinks.

- The number of soft drinks available for purchase from UK supermarkets. Here, we refer to the different options available to the consumer rather than the number of sales or the number of items available on supermarket shelves.

For the price, product size, and product diversity analyses, we stratified our results into 3 groups by sugar content: <5 g sugar per 100 mL (in which no levy applies); 5 to <8 g sugar per 100 mL (in which the lower levy rate applies); ≥8 g sugar per 100 mL (where the higher levy rate applies). Soft drinks appearing in different product sizes or in different supermarkets were included as independent observations in the study data set.

### Study design

We had no unique identifier for the soft drinks that were included in the analysis, and therefore we were not able to link all observations at different time points. Therefore, we were unable to create a panel series and structured our data set as a repeat cross-sectional design. Within this structure, we used controlled interrupted time series (CITS) analysis [29], with two intervention points: the announcement (16 March 2016) and the implementation (6 April 2018) of the SDIL. The units of analysis for the CITS were observations of all soft drinks identified from supermarkets at 85 time points between September 2015 and February 2019 (see further below). 'Soft drinks' were defined as all edible liquids (either sold ready to drink or to be reconstituted from liquid concentrates), excluding soups, alcoholic beverages (and

nonalcoholic versions), cow's milk, dried drinks (e.g., milkshake powder, instant coffee), bottled water or flavourings that need the addition of water (e.g., tea bags).

For each of the outcome measures, we conducted separate analyses on what we have called 'intervention' and 'control' drinks for brevity. 'Intervention' drinks consisted of all soft drinks except SDIL-exempt fruit juices and milk-based drinks. This set includes drinks that do not attract levy payments, because they have sugar levels below the minimum threshold of 5 g per 100 mL (e.g., 'diet' variants of popular drink brands) but represent a category into which levy-eligible drinks might fall following reformulation. 'Control' drinks consist of soft drinks that were exempt from the SDIL because of being 100% fruit juice, milk-based, or a milk alternative (regardless of sugar content). The control series was chosen because it was assumed that trends over time in this group would not be affected by the SDIL. Demonstrating this alongside effects in the intervention series would show specificity of results, strengthening the evidence that any observed relationship is causal [29].

The decision regarding how to categorise soft drinks that are neither subject to exemptions nor have sugar levels above the minimum threshold of 5 g per 100 mL is not straightforward. These drinks are not subject to the levy so could be regarded to be equivalent to drinks from exempt categories. However, we included such drinks in the intervention series as manufacturers could react to the SDIL by reducing sugar content of drinks, thereby moving drinks from categories that are taxed into categories that are not. If our study design included these non-taxed categories in the control series, then we would allow drinks to migrate from the intervention to the control series over time, which would violate our assumption that the SDIL does not affect the control series.

To report the impact of the SDIL on trends, we estimated counterfactual scenarios in which pre-SDIL trends in the variable of interest were extrapolated to simulate the likely trajectory in the absence of the SDIL, and then we estimated the difference between the observed measures from the regression models and counterfactual scenarios at 4 time points: 50 days postannouncement (5 May 2016), 50 days preimplementation (15 February 2018), 50 days postimplementation (26 May 2018), and the end of the current data set (17 February 2019, which is 317 days postimplementation). To estimate confidence intervals around the differences, we compared the 95% lower and higher confidence intervals from the observed results with point estimates from the counterfactual. The chosen timepoints for displaying results are arbitrary. The complete set of regression model results are provided in S2 Appendix allowing for estimation of results at any timepoint.

## Data

Fig 1 provides a data flowchart for the separate analyses described in this manuscript. We compiled data from 2 sources. Firstly, we used data collected from the websites of the six leading UK supermarkets (Asda, Sainsbury's, Tesco, Morrisons, Ocado, and Waitrose) that together account for 74% of UK grocery sales [30]. We collected data for this analysis using a web-scraping and data-processing software and database platform called foodDB, which has run continuously since November 2017. Full details of the methods of data collection using this tool are provided elsewhere [31]. Briefly, foodDB software collects and processes data automatically on over 99% of all food and drink products available for purchase on supermarket websites each week, including product name, nutritional information, ingredients, product size, price, and whether or not the product is on promotion. A validation exercise comparing foodDB data with equivalent data collected from 295 randomly selected products in real life stores showed high correlation between the 2 data sets for price and sugar levels and no evidence of systematic bias in comparison of the 2 data sets (S3 Appendix). The current data set

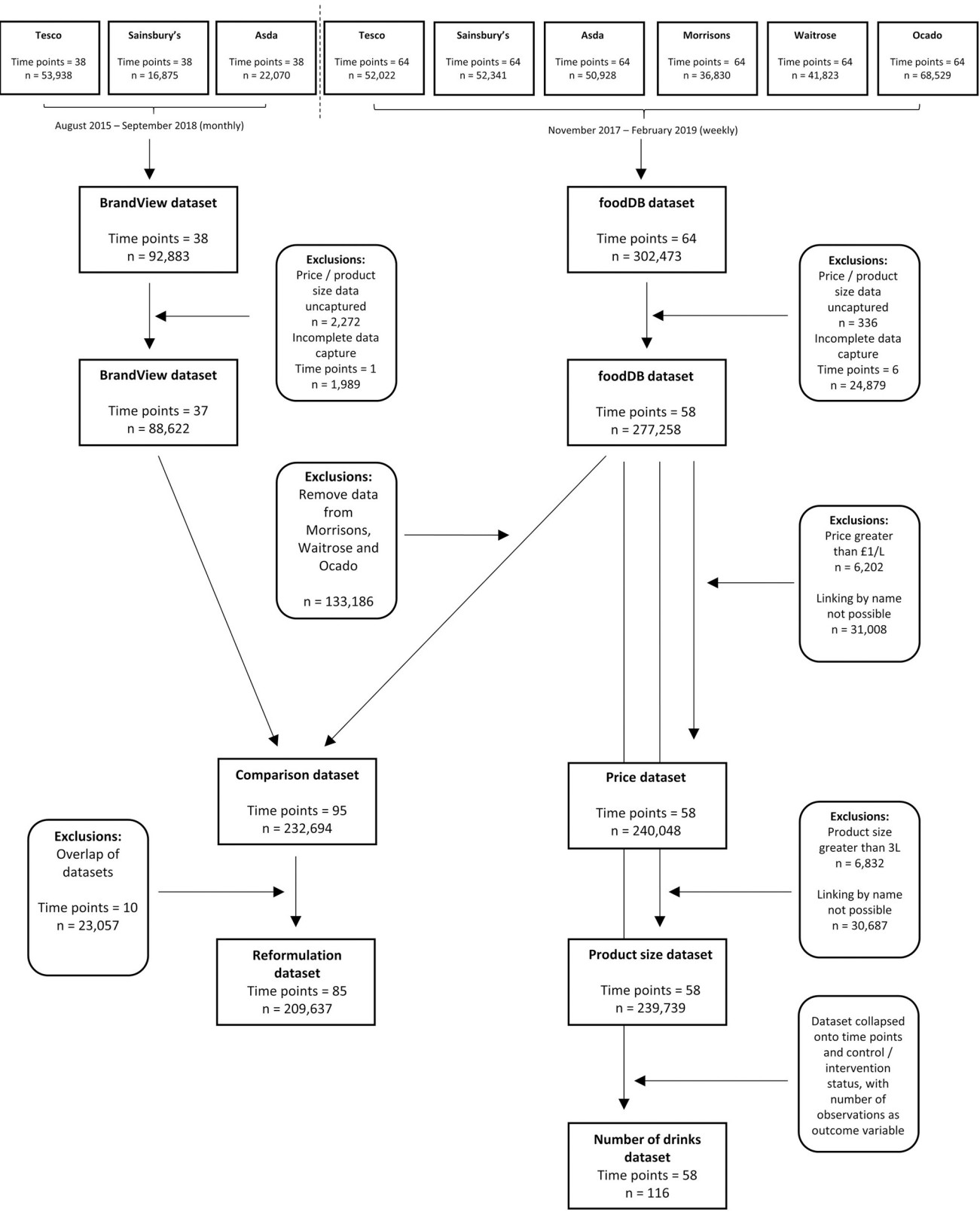

**Fig 1. Data flowchart.**

consisted of weekly data from foodDB from 26 November 2017 until 17 February 2019, consisting of 64 time points and 302,473 observations. Soft drinks were dropped from the data set if they had missing data on price or product size (there were no missing data on other study variables). Because of changes in UK supermarket website design, on some occasions the foodDB software fails to make a complete data capture. We removed these occasions from the analysis by excluding all data collected in weeks in which the total number of soft drinks collected by foodDB was less than 90% of the weekly average in the rest of the data set. After exclusions, the foodDB data set consisted of 277,258 observations over 58 time points.

The second data set provided us with data from prior to the announcement of the SDIL. We used data from 92,883 observations of soft drinks at 38 monthly time points, from 1 August 2015 to 1 September 2018 acquired from Brandview, a commercial company that collects product data using methods similar to those used in foodDB on all products available from Tesco, Sainsbury's, and Asda. After excluding observations with missing price or product size data and excluding time points in which data collection was less than 90% of average, the BrandView data set provided 88,622 observations over 37 time points (NB: the removed time point was from the first month, limiting the BrandView data set to September 2015 onwards).

We categorised all observations as 'intervention' or 'control' based on supermarket categorisation and manual inspection of product names, using equivalent methods for each data set.

## Statistical methods

We used a data-driven approach to build regression models with the aim of reproducing time trends observed in the data sets and isolating the impact of the announcement and implementation of the SDIL. We were not aiming to infer the size of the effect of a sugary drink tax on an average soft drink. This influences our modelling strategy; for example, we did not include product-level characteristics as confounding variables in the CITS models. For all outcome measures, we hypothesised that the SDIL could impact on both the level and the slope of the trend and thereby included dummy variables representing the interventions and interaction terms in our regression models (that is, a 'level and slope change analysis' [32]), and we used likelihood ratio tests to identify whether including both level and slope changes improved model fit beyond including level change alone (with a threshold for decision making of $p = 0.05$). Bernal and colleagues [29] state that 2 types of CITS model can be deployed: separate analysis of the intervention and the control series or a single model incorporating both series. The former model estimates the difference between before and after the event in the intervention series and uses the control series as a plausibility check—the event should only impact the intervention series, and effects found in the control series could be evidence of unmeasured confounding variables. The latter model estimates the difference in difference between the intervention and the control series directly. Here, we use the former approach because the population-level nature of the SDIL made it not possible to acquire location-based controls (that is, data on the same drinks but sold in supermarkets unaffected by the SDIL). For all outcome measures, regression models were run on the control drinks that included identical parameters to the equivalent models on intervention drinks. All analyses were conducted in R version 3.4.4.

For each analysis, we first observed trends in the raw data that informed the model building strategy. When nonlinear trends were observed, we included polynomial regression parameters, testing each additional parameter for improved model fit using likelihood ratio tests. Because of the very large number of possible models that could be tested, we restricted exploration of nonlinear effects only to time periods in which trends in the nonmodelled data clearly deviated from linearity. Where seasonality was observed, we included dummy variables to capture this. The specific methods used for each analysis are described below.

## Comparison of data sets

S3 Appendix describes the methods and results used to check for consistency between the foodDB and BrandView data sets. These assessments were based on a comparison data set with overlapping data from November 2017 to September 2018. To ensure comparability, all data from Waitrose, Ocado, and Morrisons were removed from the comparison data set.

## Reformulation

To conduct analyses of the impact of the SDIL on sugar content of drinks, overlapping data from BrandView were removed from the data set constructed for the comparison of the BrandView and foodDB data, resulting in a total of 209,637 observations of soft drinks from 3 supermarkets over 85 time points between September 2015 and February 2019. We built logistic regression models with dummy variables for the announcement and implementation of the SDIL.

## Price

Observation of trends in the raw data showed little evidence that the announcement of the SDIL had any impact on price of soft drinks. Therefore, the price analyses were conducted using the foodDB data set only. For the price variable, we used the price presented to the consumer for a single item purchase, which included reductions because of price promotions (for example, 10% off) but not volume-based promotions (for example, buy one get one free). We adjusted prices for an annual inflation rate of 1.7% [33], presenting all prices as of February 2019. Visual inspections of p-p plots suggested that the price variable was not normally distributed and contained a long tail of high priced drinks. To convert to normality, we first excluded outlying drinks with a price greater than £1 per litre and then log-transformed the variable. We conducted linear regression modelling on the log-transformed price variable. To protect against confounding of the results by drinks moving between SDIL tiers over time (that is, by reducing sugar content), we categorised drinks into high levy, low levy, and no levy categories on the basis of the category that they were in after the implementation of the SDIL. To do this, we matched drinks in the data set on the basis of name and excluded all drinks that could not be matched. Inspection of trends revealed that prices of soft drinks were reduced in December as Christmas promotions kicked in—we therefore included a dummy variable to indicate December in the price analyses. The price analysis data set contained 240,048 observations of soft drinks from 6 supermarkets over 58 time points.

## Product size

For the product size variable, we included drinks sold in multipacks and, for these, took the product size to be the total volume of all individual drinks in the multipack combined. For similar reasons to the price analysis, we restricted the analysis to the foodDB data set, excluded outliers and log-transformed the product size variable, and matched drinks to categorise them on the basis of levy category after implementation of the SDIL. The product size analysis data set contained 239,739 observations of soft drinks from 6 supermarkets over 58 time points.

## Number of soft drinks

For the number of soft drinks analysis, we restricted the analysis to the foodDB data set for similar reasons to the price and product size analyses. We matched the drinks by name and categorised each drink on the basis of the levy category for its last appearance in the data set. We collapsed the data set on time point and conducted linear regression analyses on the

aggregated 'number of drinks' variable. The collapse of the data set allowed us to explore whether temporal autocorrelation was present and how it affected the analyses. To do this, we included a lag term (the number of drinks at the previous time point) in the model. The number of drinks analysis consisted of 58 time points for both intervention and control drinks, with aggregated data from 6 supermarkets at each time point.

### Changes to published protocol

We made the following changes to the prespecified protocol (the work by White and colleagues [26] and reproduced in S4 Appendix). We used a different time frame for the analysis, which includes an earlier than anticipated initial date, because of our acquisition of data pre-November 2017 from BrandView. We will undertake further analyses up to the original proposed end date of April 2020 once data are available. For now we present analyses up to approximately 1 year postimplementation of the SDIL, in order to provide timely evidence of the effects of the levy. The protocol states that we will analyse the impact of the SDIL on mean sugar content of drinks—upon reflection we considered that a binary classification of the data (drinks above or below the lower levy sugar threshold) was a more appropriate way to model manufacturer response to the SDIL. The predefined analysis using mean sugar level is reported in S5 Appendix for completeness. In the protocol, we proposed using alcoholic drinks as the control series; this was altered because most alcoholic drinks do not report sugar content.

### Results

Table 1 shows descriptive statistics comparing the main outcome variables between intervention and control drinks in each data set. Further descriptive statistics for the combined BrandView and foodDB data set are available in S3 Appendix. Average sugar levels and price were higher in control drinks, but the average product size was smaller ($p < 0.001$ in all cases). There were nearly 50% more intervention than control drinks in the data sets.

Table 2 compares the proportion of drinks over the lower levy sugar threshold with the counterfactual scenario in which preannouncement trends were extrapolated, with the trend for all intervention and control drinks shown in Fig 2. The proportion of intervention drinks over the lower levy sugar threshold reduced after the announcement of the SDIL only slowly at first but with rapid changes just prior to the implementation. Just 50 days before the implementation, intervention drinks with enough sugar to be included in the levy had fallen by 19.5 (95% CI: 18.9–20.1) percentage points; 50 days after implementation intervention drinks had fallen by 30.7 (30.3–31.2) percentage points. As of February 2019, only 15.4% (14.8%–15.9%) of intervention soft drinks were above the lower levy sugar threshold. Equivalent models for the control drinks found little evidence of impact of the announcement or implementation of the SDIL on percentage of drinks above each levy threshold (see S2 Appendix for all model results). The pattern of sugar reduction in own-brand and branded drinks was very different; for own-brand drinks, sugar levels were already falling before the announcement of the SDIL, but these falls accelerated after the announcement. By the time of the implementation of the SDIL, only 6.9% (6.3%–7.6%) of own-brand intervention drinks remained over the lower levy sugar threshold and further sugar reduction stalled. For branded drinks, there was a large fall in the proportion of drinks over the lower levy sugar threshold at the point of the implementation, which resulted in a 43.5 (42.9–44.1) percentage point fall in the number of branded intervention drinks over the lower levy sugar threshold by February 2019, leaving only 17.6% (17.0%–18.2%) of branded drinks above the lower levy sugar threshold.

Table 3 shows the results of the price analysis, with Fig 3 showing the trend for intervention and control drinks, separately for branded and own-brand drinks. Branded drinks passed on

**Table 1. Descriptive statistics of sugar levels, price, product size, and number of soft drink observations.**

| Outcomes by drink category | $N^1$ | Median | IQR | $P^2$ |
|---|---|---|---|---|
| **Sugar (g per 100 mL)** | | | | |
| Higher levy tier intervention drinks | 26,755 | 10.6 | 9.8–11.6 | |
| Lower levy tier intervention drinks | 13,857 | 7.0 | 6.3–7.5 | |
| No levy tier intervention drinks | 92,837 | 0.5 | 0.0–4.3 | |
| All intervention drinks | 133,449 | 4.2 | 0.2–7.1 | |
| All control drinks | 76,188 | 8.2 | 3.4–10.0 | <0.001 |
| **Price (p per 100 mL)[3]** | | | | |
| Higher levy tier intervention drinks | 12,813 | 25.4 | 20.2–36.5 | |
| Lower levy tier intervention drinks | 12,535 | 33.8 | 26.9–40.7 | |
| No levy tier intervention drinks | 111,626 | 14.2 | 9.0–24.0 | |
| All intervention drinks | 136,974 | 17.3 | 10.1–27.4 | |
| All control drinks | 103,074 | 21.3 | 14.3–37.5 | <0.001 |
| **Product size (mL)** | | | | |
| Higher levy tier intervention drinks | 12,111 | 750 | 497–1,006 | |
| Lower levy tier intervention drinks | 12,613 | 749 | 500–781 | |
| No levy tier intervention drinks | 109,726 | 1,000 | 548–1,974 | |
| All intervention drinks | 134,450 | 1,000 | 500–1,842 | |
| All control drinks | 105,289 | 950 | 593–1,000 | <0.001 |
| **Number per week** | | | | |
| Higher levy tier intervention drinks | 58 | 256 | 252–291 | |
| Lower levy tier intervention drinks | 58 | 298 | 287–311 | |
| No levy tier intervention drinks | 58 | 2,274 | 2,245–2,319 | |
| All intervention drinks | 58 | 2,862 | 2,795–2,902 | |
| All control drinks | 58 | 1,971 | 1,946–2,010 | <0.001 |

[1]For 'sugar', 'price', and 'product size', this represents the total number of observations over all time points included in the analyses. For 'number per week', all observations are collapsed in each time point, so this represents the number of time points in the analyses.

[2]From Wilcoxon rank sum test comparing intervention and control drinks.

[3]Adjusted to February 2019 prices. Note that for price and product size, the categorisation by levy tier is based on the categorisation of products after implementation of the levy, for number per week it is based on the last observation in the data set, and for sugar it is based on the sugar level at the point of observation.

**Abbreviations:** IQR, interquartile range

**Table 2. Difference between observed and counterfactual (extrapolation of preannouncement trends) percentage of soft drinks over the lower levy sugar threshold.**

| Drink categories | Percentage over lower levy threshold before announcement | Difference in percentage[1] of drinks over lower levy sugar threshold (95% confidence intervals) | | | |
|---|---|---|---|---|---|
| | | 5 May 2016 (50 days postannouncement) | 15 February 2018 (50 days preimplementation) | 26 May 2018 (50 days postimplementation) | 17 February 2019 (end of data set) |
| **All intervention drinks** | 51.7 (50.9–52.6) | −0.1 (−1.3 to 1.1) | −19.5 (−20.1 to −18.9) | −30.7 (−31.2 to −30.3) | −33.8 (−34.4 to −33.3) |
| Branded intervention drinks | 57.9 (57.0–59.0) | −1.1 (−2.4 to 0.3) | −23.8 (−24.5 to −23.1) | −38.3 (−38.9 to −37.8) | −43.5 (−44.1 to −42.9) |
| Own-brand intervention drinks | 34.8 (33.2–36.4) | 2.5 (0.3–4.7) | −11.5 (−12.2 to −10.7) | −12.2 (−12.9 to −11.5) | −9.4 (−10.2 to −8.6) |
| **All control drinks** | 68.1 (66.8–69.3) | 0.6 (−1.0 to 2.2) | −5.8 (−6.6 to −5.1) | −6.9 (−7.6 to −6.2) | −7.9 (−8.9 to −7.0) |

[1] Results are presented as percentage point differences compared to the counterfactual (extrapolation of preannouncement trend).

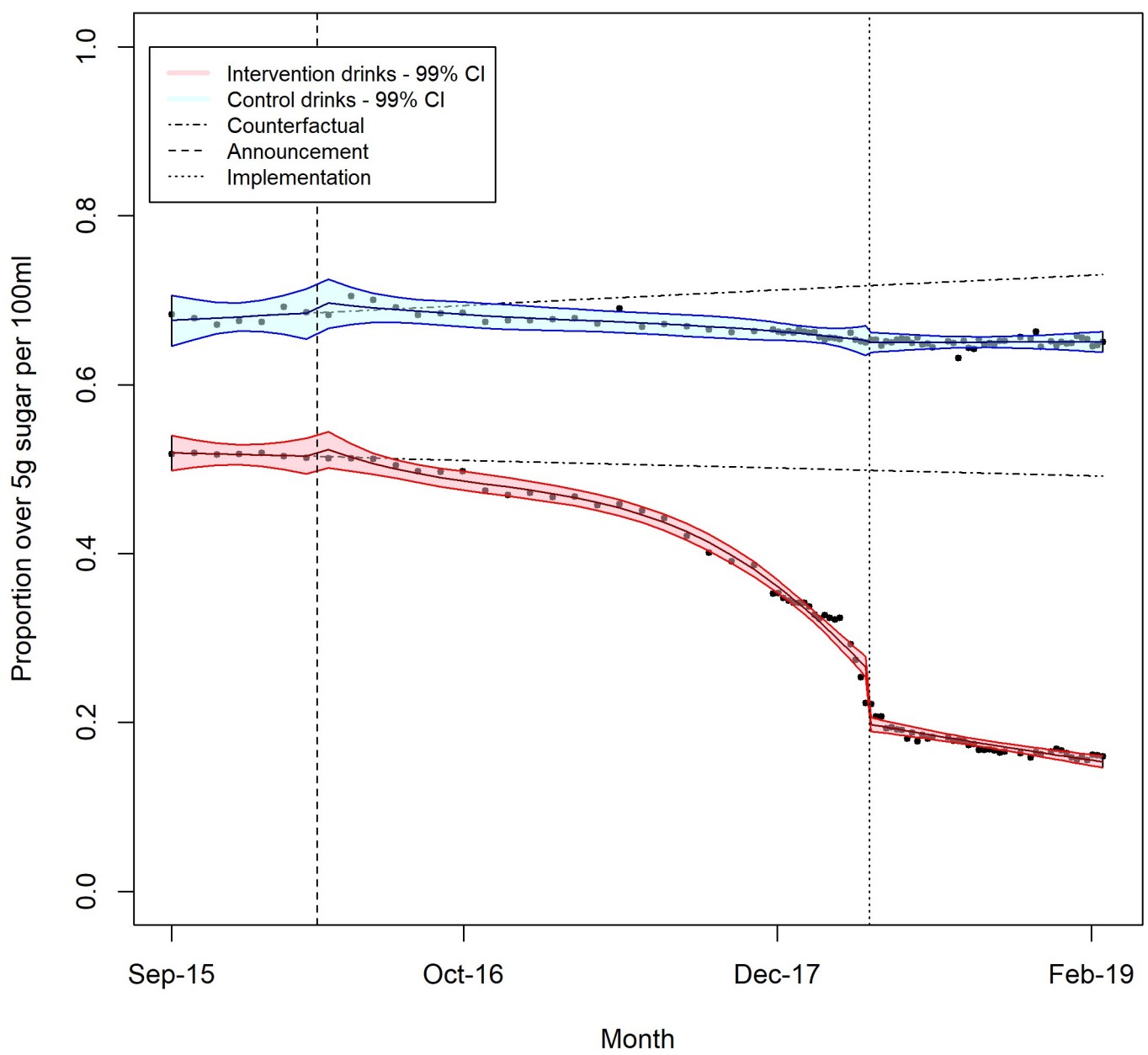

**Fig 2. Proportion of soft drinks over the lower levy sugar threshold.**

about half of the levy on higher levy tier drinks (that is, the price increase on these drinks was half of the levy rate), whereas the prices of lower levy tier drinks reduced after implementation of the SDIL. In contrast, own-brand drinks saw large changes in price with higher levy tier drinks reducing in price by 62.5 p per L (52.4–72.1) and lower levy tier drinks increasing by 68.6 p per L (56.9–81.1); Fig 2 shows how the price point for these 2 categories converged after the implementation of the SDIL.

Table 4 shows the results for product size and number of drinks available in supermarkets. For product size, there was very little impact of the SDIL on branded drinks, which showed only small fluctuations in product size after implementation of the SDIL of similar magnitude

**Table 3. Difference between the observed and counterfactual (extrapolation of preimplementation trends) in prices of soft drinks as of 26 May 2018 (50 days postimplementation).**

| Drink categories | Mean price before implementation, pence (p) per litre (95% CI)[1] | Difference in price, pence (p) per litre (95% CI)[1] | Pass-on rate[2] |
|---|---|---|---|
| **All drinks** | | | |
| Higher levy tier intervention drinks | 251.0 (240.3–262.2) | 7.5 (3.7–11.5) | 31% (15%–48%) |
| Lower levy tier intervention drinks | 319.3 (305.8–333.4) | −10.7 (−15.3 to −6.0) | −59% (−85% to −33%) |
| No levy tier intervention drinks | 135.4 (127.7–143.6) | 3.6 (2.6–4.7) | n/a |
| Control drinks | 227.5 (215.7–239.9) | −1.5 (−3.0 to 0.1) | n/a |
| **Branded drinks** | | | |
| Higher levy tier intervention drinks | 250.5 (239.7–261.8) | 11.8 (7.7–15.9) | 49% (32%–66%) |
| Lower levy tier intervention drinks | 336.5 (323.6–350.0) | −17.4 (−22.0 to −12.8) | −97% (−122% to −71%) |
| No levy tier intervention drinks | 162.9 (154.9–171.4) | 2.6 (1.4–3.8) | n/a |
| Control drinks | 269.3 (256.6–282.6) | −4.1 (−5.9 to −2.2) | n/a |
| **Own-brand drinks** | | | |
| Higher levy tier intervention drinks | 268.8 (260.8–277.1) | −62.5 (−72.1 to −52.4) | −260% (−300% to −218%) |
| Lower levy tier intervention drinks | 123.2 (118.8–127.8) | 68.6 (56.9 to 81.1) | 381% (316%–451%) |
| No levy tier intervention drinks | 70.7 (67.1–74.5) | −0.8 (−1.9 to −0.3) | n/a |
| Control drinks | 122.8 (118.6–127.1) | 0.1 (−1.1 to 1.4) | n/a |

[1]Adjusted to February 2019 prices.

[2] Higher levy tier drinks are levied at £0.24 (24 p) per litre; lower levy tier drinks are levied at £0.18 (18 p) per litre; no levy tier drinks and control drinks are not levied. The pass-on rate is the percentage of the levy that was passed to the consumer as a change in price.

to variations observed in the control drinks. However, for own-brand drinks, we observed a similar convergence as seen in the price analyses; here, drinks levied at the lower level reduced in average product size, and drinks levied at the higher rate increased until the average product size in both were similar. For product diversity, the inclusion of lag terms had little impact on model results. The models used for Table 4 and reported in S2 Appendix did not account for autocorrelation. We saw little evidence that the SDIL impacted on the number of drinks available in supermarkets; in general, products that left were replaced with new products. The largest difference between the observed and counterfactual scenarios was for control drinks, and these results were based on regression models that suggested only very weak evidence of impact of the SDIL (see S2 Appendix).

## Discussion

The SDIL was associated with a large reduction in the percentage of soft drinks (particularly branded drinks) that are subject to the levy because of large reductions in the sugar levels of these drinks. There was no evidence for similar reductions in control SDIL-exempt drinks, suggesting that the SDIL was the motivating factor for this change. We found that the levy was not directly passed on to the consumer through commensurate increases in the prices of targeted drinks, but manufacturers and retailers appear to have taken the opportunity to undertake wider revision of their entire soft drink market offer. For example, there were changes in both prices and volumes of drinks; only half of the levy on branded higher levy tier drinks was passed on to consumers, whereas low sugar variants also increased in price, and price points for own-brand higher and lower levy tier drinks converged. Without sales data to weight the results reported here, it is not possible to estimate whether the full extent of the levy was passed on to consumers via increases in prices. Our analysis of product size suggested that manufacturers of branded drinks did not react to the SDIL by changing product sizes.

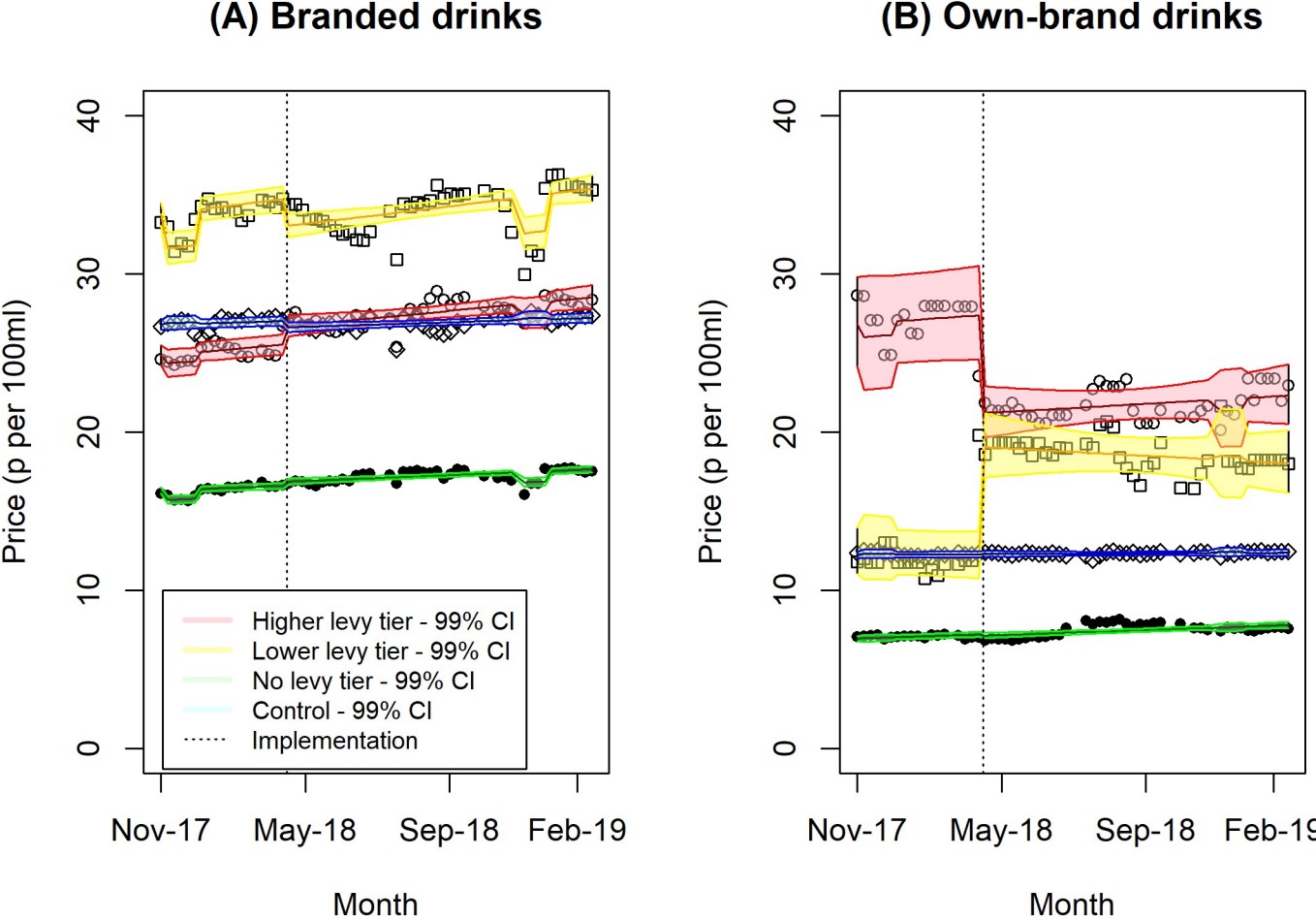

**(A) Branded drinks**

**(B) Own-brand drinks**

Legend:
- Higher levy tier - 99% CI
- Lower levy tier - 99% CI
- No levy tier - 99% CI
- Control - 99% CI
- Implementation

**Fig 3. Change in price of (A) branded and (B) own-brand soft drinks by sugar content.**

However, supermarkets made large changes to their own-brand product sizes of higher and lower levy tier drinks. About 30% of the price per volume increase on own-brand lower levy tier drinks can be accounted for by the reduction in product sizes—an instance of so-called 'shrinkflation' [34]. We did not observe any changes in the number of soft drinks available to consumers as a result of the SDIL.

These results suggest that the SDIL has stimulated decreases of sugar levels of soft drinks. Reductions were because of reformulation of existing products and replacement of drinks with lower sugar varieties. The stimulus for these changes are likely to include both supply and demand factors—manufacturers may be influenced to reduce sugar levels to avoid the levy or may be prompted by a change in demand for lower sugar soft drinks after the widespread media attention related to the announcement of the levy. Our results also confirm that the SDIL currently only applies to a small percentage of the soft drinks that are available in the UK grocery market; control drinks make up over a third of the available soft drinks, and, by February 2019, only 15% of the intervention drinks were being levied (the remaining 85% had sugar levels lower than the levy sugar threshold). The lower levy sugar threshold (5 g per 100 mL) is

**Table 4. Difference between product size and diversity in product range of soft drinks in the modelled and counterfactual (extrapolation of preimplementation trends) results as of 26 May 2018 (50 days postimplementation).**

| Drink categories | Difference in product size, mL (95% CI) | Difference in number of products available (95% CI) |
|---|---:|---:|
| **All drinks** | | |
| Higher levy tier intervention drinks | 1 (−15 to 17) | −3 (−12 to 6) |
| Lower levy tier intervention drinks | 13 (3–23) | −1 (−11 to 8) |
| No levy tier intervention drinks | −2 (−10 to 6) | −54 (−120 to11) |
| Control drinks | 4 (0–8) | −111 (−161 to −61) |
| **Branded drinks** | | |
| Higher levy tier intervention drinks | −7 (−23 to 11) | −10 (−18 to −1) |
| Lower levy tier intervention drinks | 16 (6 to 27) | 2 (−7 to 10) |
| No levy tier intervention drinks | 0 (−9 to 9) | −13 (−63 to 38) |
| Control drinks | 6 (1–11) | −91 (−131 to −51) |
| **Own-brand drinks** | | |
| Higher levy tier intervention drinks | 172 (133–214) | 6 (5–7) |
| Lower levy tier intervention drinks | −141 (-170 to −111) | 2 (1–4) |
| No levy tier intervention drinks | 6 (−7 to 20) | −42 (−59 to −24) |
| Control drinks | 7 (−0 to 15) | −20 (−32 to −8) |

set at a higher level than for the majority of jurisdictions that have instituted sugar drink taxes worldwide [35], and our data show that in February 2019, 65% of control drinks contained ≥5 g sugar per 100mL. After the implementation of the SDIL, we observed a peak in the proportion of intervention drinks with a sugar level between 4.5 and 5.0 g per 100 mL (see S5 Appendix), suggesting that many manufacturers chose to reformulate to just below this threshold. The second chapter of the UK Government's childhood obesity plan [36] suggests that the SDIL may be extended to milk-based drinks. Our analyses suggest that if manufacturers of milk-based drinks behave similarly, then this extension could prompt reductions in sugar levels. Given the preponderance of drinks with sugar levels just below 5 g per 100 mL, a gradual lowering of the lower levy sugar threshold, similar to gradual lowering of salt targets in the UK [37], could also have public health benefits. We also observed that the SDIL was associated with increases in price of nontargeted drinks (intervention drinks with sugar levels lower than the lower levy sugar threshold, such as diet variants). This has not previously been observed for other sugary drink taxes implemented elsewhere [22, 24, 25, 38], suggesting that the nature of the levy (a levy on manufacturers and importers based on reported sales, rather than an excise tax on consumers) may have influenced industry behaviour more widely.

The tiered design of the SDIL is also being implemented in other jurisdictions, including South Africa, Ireland, and Portugal [35], and it is therefore important to establish whether such a design influences the behaviour of manufacturers. We analysed a comprehensive set of data on soft drinks available for purchase in the leading supermarkets in the UK, which provided adequate statistical power for the analyses and generalisability of the results to the UK grocery market. However, because of the nonrandomised design of the study, it is not possible

to rule out the possibility of residual confounding in our analyses. We have demonstrated specificity for some of our results—similar changes in sugar content, price, and product size were not shown in the control drinks—which suggests that the results were not confounded by unmeasured variables.

Our results are not sales weighted, so they do not give an account of how sugar consumption from drinks may have changed over the time period. We have not been able to include soft drinks that are only available in supermarket chains or other types of retail outlet outside of those included in this analysis; although, because the supermarkets included here are the market leaders, this is unlikely to be a major limitation. We were not able to identify soft drinks produced or distributed by manufacturers and importers with UK sales less than 1 million litres per year, which were therefore incorrectly included in 'intervention' drinks. Data collected from web scraping tools (which is the case for both data sets used in these analyses) only reflect data that are presented in online supermarkets, which may not reflect the in-store environment, although our initial validation exercise on 295 food and drink products show no evidence of systematic bias when collecting data from online supermarkets (S3 Appendix). The data-driven approaches that we have used for the modelling strategy may lead to overfitted models, which can limit the generalisability of these results to other jurisdictions considering introducing a similarly structured levy [39]. Further, our aim was to reproduce trends observed in the UK over the time period studied using a near-comprehensive data set of drinks available for purchase, but we did not aim to isolate the independent effect of the SDIL on an 'average' drink adjusted for product and supermarket characteristics. As a result, it is unlikely that the magnitude of our results will be generalizable to other jurisdictions considering introducing a similar levy. The control series may not be isolated from effects of the SDIL (for example, manufacturers may choose to adapt prices of control drinks in response to the SDIL because they are a potential substitute for intervention drinks). Because of the lack of a unique product identifier in the data set, it was not possible to analyse these data as a panel series, and hence we were unable to account for the autocorrelation structure in any of the analyses with the exception of the 'number of products' analysis.

Other studies have used CITS to evaluate the impact of voluntary soft drink price increases that have been implemented in the UK [40,41] and soft drink taxes implemented elsewhere in the world [23,24,25, 38] and have shown that they have resulted in reduced sales of targeted drinks [42] and that price increases are generally passed on to the consumer on targeted drinks but not always the full tax; the French soda tax had a differential pass-on rate in different communities, with more deprived areas having large pass-on rates and an average pass-on rate of 40% [38]. To our knowledge, no previous study has evaluated the impact of an economic instrument for stimulating reformulation of soft drinks. A public health campaign to encourage voluntary soft drink reformulation in Austria was shown to result in a 13% increase in the number of drinks under the campaign threshold of 7.4 g sugar per 100 mL over a 7 year period [43], and the voluntary UK salt reduction campaign that began in the mid-2000s has been shown to have reduced salt levels in commonly consumed food groups by 7% between 2006 and 2011 [44] and up to 47% since 2004 for breakfast cereals (albeit based on a small sample) [45]. An evaluation of the UK Public Health Responsibility Deal, which asked food manufacturers to make pledges for reformulation, found that inherent conflicts within the food system limit the ability of voluntary processes to make sizeable impacts [46]. Our results show a much steeper decline in targeted nutrient levels than those that have been observed in the UK and elsewhere, suggesting that economic instruments may be more effective at changing manufacturer behaviour than voluntary public health interventions. Public Health England (PHE) used data provided by a commercial party on sales of soft drinks between 2015 and 2018 and found that there was reduction of 29% in sales-weighted average sugar content of drinks over this

time period [47]. A separate analysis found a 30% reduction in sales-weighted sugar levels between 2015 and 2018 [48] using data sets independent from PHE. The PHE analysis differs from ours in 3 important aspects: they do not account for background trends in sugar levels, their data includes purchases from a wider range of retail outlets, and their results are sales-weighted. Our equivalent analysis is shown in S5 Appendix; we found a 2.13 g per 100 mL (2.08–2.18) fall in sugar levels in intervention drinks because of the announcement and implementation of the SDIL; this relates to a 38% reduction from average sugar levels in September through December 2015.

The SDIL incentivised many manufacturers to reduce sugar in soft drinks. Some of the SDIL was passed onto consumers as higher prices but not always on targeted drinks. These changes could reduce population exposure to sugars and associated health risks. Further work should investigate the impact of the SDIL on consumer behaviour by influencing purchasing and consumption of soft drinks, as has been shown elsewhere in the world [23–25, 49]. The impact of these changes on consumer behaviour, including substitution effects, will be explored as part of our ongoing evaluation of the SDIL, which will also explore the impact of the SDIL on the economy, consumer attitudes, measured short term and modelled long term health outcomes [26].

## Supporting information

**S1 Appendix. Analysis of impact of soft drinks industry levy on proportion of drinks over higher levy threshold (8 g sugar per 100 mL).**
(DOCX)

**S2 Appendix. Model parameters for all models presented in the main analysis and supplementary material.**
(DOCX)

**S3 Appendix. Comparison of foodDB and BrandView data sets.**
(DOCX)

**S4 Appendix. Prepublished protocol.**
(DOCX)

**S5 Appendix. Analysis of impact of soft drinks industry levy on mean sugar levels.**
(DOCX)

## Acknowledgments

We thank Richard Smith, David Pell, and Harry Rutter for helpful comments on the manuscript. The data used in this paper were obtained from the websites of Tesco, Sainsbury's, Morrisons, Asda, Ocado, and Waitrose supermarkets in the UK and in part from Brandview Ltd. Because of UK copyright law, the data used in this analysis can only be shared under licence ensuring that they will only be used for noncommercial purposes. Please see the data availability statement for details.

The views expressed are those of the authors and not necessarily those of the NHS, the NIHR, or the Department of Health and Social Care.

## Author Contributions

**Conceptualization:** Peter Scarborough, Richard A. Harrington, Adam Briggs, Mike Rayner, Jean Adams, Steven Cummins, Martin White.

**Data curation:** Vyas Adhikari, Richard A. Harrington, Ahmed Elhussein.

**Formal analysis:** Peter Scarborough, Vyas Adhikari, Richard A. Harrington.

**Funding acquisition:** Peter Scarborough, Richard A. Harrington, Adam Briggs, Mike Rayner, Jean Adams, Steven Cummins, Martin White.

**Investigation:** Peter Scarborough, Vyas Adhikari, Richard A. Harrington, Ahmed Elhussein.

**Methodology:** Peter Scarborough, Vyas Adhikari, Richard A. Harrington, Adam Briggs, Mike Rayner, Jean Adams, Steven Cummins, Tarra Penney, Martin White.

**Project administration:** Peter Scarborough, Tarra Penney, Martin White.

**Software:** Richard A. Harrington.

**Supervision:** Mike Rayner.

**Writing – original draft:** Peter Scarborough, Ahmed Elhussein, Adam Briggs.

**Writing – review & editing:** Peter Scarborough, Vyas Adhikari, Richard A. Harrington, Ahmed Elhussein, Adam Briggs, Mike Rayner, Jean Adams, Steven Cummins, Tarra Penney, Martin White.

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
