## [Decision Letter · Decision Letter 0]

29 Aug 2019

Dear Dr. Scarborough,

Thank you very much for submitting your manuscript "Impact of the announcement and implementation of the UK Soft Drinks Industry Levy on sugar content, price, product size and number of available soft drinks in the UK, 2015-19: controlled interrupted time series analysis" (PMEDICINE-D-19-02557) for consideration at PLOS Medicine. 

[LINK]

In light of these reviews, I am afraid that we will not be able to accept the manuscript for publication in the journal in its current form, but we would like to consider a revised version that addresses the reviewers' and editors' comments. Obviously we cannot make any decision about publication until we have seen the revised manuscript and your response, and we plan to seek re-review by one or more of the reviewers. 

We expect to receive your revised manuscript by Sep 19 2019 11:59PM. Please email us (plosmedicine@plos.org) if you have any questions or concerns.

We look forward to receiving your revised manuscript. 

Sincerely,

Adya Misra, 

Senior Editor 

PLOS Medicine

plosmedicine.org

If a prospective analysis plan (from your funding proposal, IRB or other ethics committee submission, study protocol, or other planning document written before analyzing the data) was used in designing the study, please include the relevant prospectively written document with your revised manuscript as a Supporting Information file to be published alongside your study, and cite it in the Methods section. A legend for this file should be included at the end of your manuscript. 

If no such document exists, please make sure that the Methods section transparently describes when analyses were planned, and when/why any data-driven changes to analyses took place. 

In either case, changes in the analysis—including those made in response to peer review comments—should be identified as such in the Methods section of the paper, with rationale.

Abstract- Please explain what is meant by “209,637 observations of soft drinks at 85 time points”

Abstract-Please consider introducing the terms lower levy, higher levy and no levy categories earlier in the abstract

Abstract-Please quantify the main results (with 95% CIs and p values)

Abstract-In the last sentence of the Methods and Findings section, please describe the main limitation(s) of the study's methodology

DAS- Note that a study author cannot be the contact person for the data. Please deposit your data with a third party such as a research ethics committee or data committee for data access by third parties and provide this information within the data availability statement

Introduction-Line 82 please introduce the NHS on first view

Page 9 Line 189-190 please clarify why these time points were excluded from analysis

Line 221- please clarify what is meant by reasonably normally distributed and if linear regression models were appropriately used 

Line 360 Please clarify what is meant by “obvious changes in the number of soft drinks available” and clarify in the methods section how this was evaluated 

Line 363- Please clarify if the speed of reformulation was directly tested in this study or consider toning down this sentence

Line 385-Please alter all assertions of primacy to “to the best of our knowledge”

Your study is observational and therefore causality cannot be inferred. Please remove language that implies causality, such as [ ]. Refer to associations instead

Comments from the reviewers:

Reviewer #1: See attachment

Michael Dewey

Reviewer #2: This very well-written paper describes changes in soft drink sugar content, product size, and availability in the UK after the announcement and implementation of the UK SDIL. The findings make an important contribution to the field of public health and are particularly important as the first evidence of the impact of tiered taxes on beverage industry behaviors. 

I have only minor suggestions for revisions.

Introduction: 

Page 4 - line 92: Many SSB taxes globally ,and all SSB taxes passed recently in the U.S. (Philadelphia, Berkeley, Oakland, San Francisco, Seattle, West Virginia, Boulder) are excise taxes levied on quantity - not ad valorem taxes. These recent taxes are expected to raise shelf prices, but also aim to generate revenue for local jurisdictions. The important distinction seems to be the SDIL's attempt to encourage product reformulation.

Methods:

Include the number of unique products available in methods - I assume observations represents the total observations but it's not clear how many unique products this represents

Lines 211-12: clarify that proportion over levy threshold limited to smaller (Brandview) dataset (I believe this is the case)

Lines 212-217: clarify if "single item" is equivalent to non-multipack

Lines 222-224: clarify if clustering by unique product was accounted for in models and provide rationale for approach

Results:

Table 1: explain "Number per week"- not clear what this represents (I assume unique products but N is then confusing)

Line 282: provide baseline % above threshold in text

Table 2: include baseline % above threshold

Lines 311-314 and Table 3: It would be interesting to know how price changed among lower levy tier drinks that did not change category over time (if not different from results presented, would be useful to simply state that). It seems that results as presented would reflect reduction in price related to moving from higher to lower levy tier.

Appendix:

For S2, in text, provide "baseline" mean sugar content among eligible soft-drinks

Figure S4 vs. text: text states mean sugar content was 2.13 in Feb 2019 but in figure final mean sugar content appears to be above 3 g/100ml. Clarify discrepancy.

Reviewer #3: Review of PLOS Medicine manuscript PMEDICINE-D-19-02557 titled "Impact of the announcement and implementation of the UK Soft Drinks Industry Levy on sugar content, price, product size and number of available soft drinks in the UK, 2015-19: controlled interrupted time series analysis."

This study examines the impact of the U.K. Soft Drinks Industry Levy (SDIL), a tiered beverage tax based on sugar content, on tax pass-through, sugar content of beverages with respect to the levy thresholds, product size of available soft drinks, and availability of soft drinks. The authors draw on two different data sources that provide information on food and beverage products available for purchase on supermarket websites each week. The study uses an interrupted time series approach. The authors do not have a control site; however, they do make comparisons to a counterfactual based on the extrapolation of pre-announcement trends. It is critically important to understand the extent to which beverage taxes are passed on to consumers in the form of higher prices, the extent to which tiered taxes can incentivize reformulation, and the extent that the supply of beverages in the market place changes post-tax; hence, this study addresses important questions. However, the contribution of this study is severely limited by a number of queries and concerns related to the data and the lack of rigor in the specification an estimation of the empirical models which leads to significant distrust in the validity and interpretation of the results. 

1. Page 4, line 92. The paper states that soft drink taxes introduced elsewhere are normally valorem taxes. This is not true. Many are specific excise taxes levied on a per unit basis. See the following UNC website which provides an overview of the different types of beverage taxes worldwide: https://www.dropbox.com/s/bqbj501wgocor24/UNCGFRP_SSB_tax_maps.pdf?dl=0

2. Page 5, lines 105-106. The authors state that no soft drink tax that incentivizes industry to reduce sugar content - i.e., tired taxes - have been evaluated. There are at least two published evaluations of Chile's effective tiered tax.

3. Page 6, line 131. The authors often refer to the "number" of soft drinks of available. The number may be confused with extent or depth of stock available which is not what is being assessed. Please be careful on wording.

4. Page 7, lines 154-156. The authors note that the controls were chosen because it was assumed that they would not be affected. But they are substitutes and hence will have cross-price effects. 

5. The foodDB data set consists of weekly data from November 2017 to February 2019. This does not even provide one full year of pre-tax data. There will be issues of seasonality. This is not at all addressed in the methods section. The analyses for price, product size, and availability of products was limited to this data set.

6. There were inadequate descriptives given on the data. How many unique products? What about classifications by drink type - soda, energy drinks etc. 

7. The methods do not provide any discussion on how the samples were constructed and whether they were balanced over time. Related to this is a significant concern about the fact that as beverages are reformulated over time they will actually jump across tax tiers. Thus, there is a concern about the composition of drinks changing within tiers over time. Much of what you are reporting say for tax pass-through by levy category may end up being an artifact of changing composition rather than within product effects. How do results change when the samples are balanced? 

8. Related to the point above -- how do results change when product fixed effects are used? What product characteristics are controlled for in the regressions (it appears none from the appendices)? How is seasonality addressed? Do you include controls which supermarket the observation comes from? This is all critical to the model specification and not at all dealt with in the paper. 

9. Also of concern is the fact that interrupted time series analysis requires identification of the autocorrelation process of the time series. What autocorrelation structure did the authors use and how was this tested? There is no mention of this in the paper.

[LINK]

---

## [Decision Letter · Decision Letter 1]

8 Dec 2019

Dear Dr. Scarborough,

Thank you very much for re-submitting your manuscript "Impact of the announcement and implementation of the UK Soft Drinks Industry Levy on sugar content, price, product size and number of available soft drinks in the UK, 2015-19: controlled interrupted time series analysis[ISRCTN 18042742]" (PMEDICINE-D-19-02557R1) for review by PLOS Medicine.

I have discussed the paper with my colleagues and the academic editor and it was also seen again by two reviewers. I am pleased to say that provided the remaining editorial and production issues are dealt with we are planning to accept the paper for publication in the journal. The Academic Editor has provided additional comments for authors to consider and incorporate into their submission to provide better context to the work. 

[LINK]

We look forward to receiving the revised manuscript by Dec 13 2019 11:59PM. 

Sincerely,

Adya Misra, PhD

Senior Editor 

PLOS Medicine

plosmedicine.org

Comments from Academic Editor

They are looking at changes induced by consumer demand and industry reformulation/creation of the new products. And the results accelerated after the SDOL was announced I think they must understand this context better and not ascribe changes just to industry shifts. These are very complex and consumer demand plays a huge role in all of this. 

they are combining supply and consumer demand issues and calling them reformulation. Reformulation means looking at the identical product and seeing how it changes in composition. They conflate the demand shifts that come with all the media and the price changes with the industry shifts in product composition and that is again also linked with new products. And the product size changes result from both reformulation and the price effects on demand getting them to try to cut size and keep the prices closer to the earlier prices.

Requests from Editors:

Title- please remove ISRCTN registry details and revise to "..a controlled interrupted time series analysis"

Abstract- in the methods and findings section you say “SDIL was associated with a small impact on… and no impact on …” could you please add a p value/95% CI or percentage points as appropriate. The same goes for “sizeable differences in outcomes for branded and own brand drinks” 

Data availability statement- please include the data availability statement as discussed, in adherence to PLOS Data policy. Please also revise the acknowledgements with regards to data as appropriate

Author summary- would it be more appropriate to say “encourage the soft drinks industry” rather than “influence” as per SDIL wording in the policy documents. Please also include the types of drinks excluded from the SDIL. 

Author summary- I believe it is overreaching to say “very large changes to sugar levels in drinks” and “large impact on the soft drinks industry” based on the results

Line 467- please avoid assertions of primacy 

Line 539 – data available from author – remove and update as discussed re Data Policy

In the ref list please remove the square brackets

Please discuss the NHE report on sugar intake that came out this September or August and compare your findings with it.

Please provide details regarding related manuscript(s) under review elsewhere, mentioning how these works differ and illustrate any overlap between various manuscripts, if any. 

All figures- please ensure the key matches the data in the graphs. For example Figure 3 the key indicates red lines for data but the graph contains a different version of red appearing pink. Is it possible to keep this consistent? 

Comments from Reviewers:

Reviewer #1: The authors have addressed my comments in their rebuttal and it is now much clearer what they did but I still find myself baffled by some of the decisions about the classification of beverages. This may be because I am more familiar with the analysis of clinical data-sets rather than economic analyses, I know custom and practice varies across disciplines. I leave this issue to the other reviewers who have more knowledge of this topic area.

One thing which still concerns me is the data-driven nature of the analysis. The authors' rebuttal and edits disclaim any intent to have a generalisable model and instead focus on having as good a fit as possible to the UK data-set. This seems to me rather disingenuous as they do also want to suggest that there are lessons potentially to be learned by other jurisdictions which implies generalisability.

Michael Dewey

Reviewer #3: Review of PLOS Medicine manuscript PMEDICINE-D-19-02557R1 titled "Impact of the announcement and implementation of the UK Soft Drinks Industry Levy on sugar content, price, product size and number of available soft drinks in the UK, 2015-19: controlled interrupted time series analysis."

The revised version of this study is much improved. However, there are still a number of revisions needed prior to publication.

1. There is no valid reason not to provide baseline data. In fact, it is customary practice in empirical work. The authors somehow think that the readers will confuse this with the counterfactual. Please have some faith in the basic intelligence level of your readers.

2. Please show descriptive statistics by tier in Table 1. 

3. Please show mean by tier in Table 3 and show means by branded versus own-brand to help readers interpret the results.

4. Why are you only assessing proportion above 5g? A key point of interest is the extent of change in high-sugar beverages.

5. The author summary section uses causal language ("The percentage of drinks with sugar over 5g … fell ... because of the SDIL." The editors cautioned you not to attribute causation. Another causal reference is made on page 8.

6. P. 14. How many observations did you lose excluding drinks with price greater than £1 per litre. This would disproportionately impact certain types of drinks such as energy drinks which are more expensive. 

7. The author assumption that the SDIL does not affect the control series is simply not valid for some of the outcomes. True you do not measure consumer purchasing and so there will not be cross-price effects to consider in this study but you do measure price itself for instance and that could be impacted (i.e., substitution could push up price of untaxed beverages) as could be sizes, numbers available. This is critical to your study and still not justified.

[LINK]

---

## [Editor Report · Decision Letter 2]

7 Jan 2020

Dear Dr Scarborough, 

On behalf of my colleagues and the academic editor, Dr. Barry M. Popkin, I am delighted to inform you that your manuscript entitled "Impact of the announcement and implementation of the UK Soft Drinks Industry Levy on sugar content, price, product size and number of available soft drinks in the UK, 2015-19: a controlled interrupted time series analysis" (PMEDICINE-D-19-02557R2) has been accepted for publication in PLOS Medicine. 

PRODUCTION PROCESS

PRESS

PROFILE INFORMATION

Thank you again for submitting the manuscript to PLOS Medicine. We look forward to publishing it. 

Best wishes, 

Adya Misra, PhD

Senior Editor 

PLOS Medicine

plosmedicine.org